# Selective Pre-training for Private Fine-tuning

**Da Yu**  *yuda3@mail2.sysu.edu.cn*
*Sun Yat-sen University*

**Sivakanth Gopi**  *sigopi@microsoft.com*
*Microsoft Research*

**Janardhan Kulkarni**  *jakul@microsoft.com*
*Microsoft Research*

**Zinan Lin**  *zinanlin@microsoft.com*
*Microsoft Research*

**Saurabh Naik**  *snaik@microsoft.com*
*Microsoft*

**Tomasz Lukasz Religa**  *toreli@microsoft.com*
*Microsoft*

**Jian Yin**  *issjyin@mail.sysu.edu.cn*
*Sun Yat-sen University*

**Huishuai Zhang**  *zhanghuishuai@pku.edu.cn*
*Peking University*

**Reviewed on OpenReview:** *https://openreview.net/forum?id=y3u8OpPHxz*

## Abstract

Text prediction models, when used in applications like email clients or word processors, must protect user data privacy and adhere to model size constraints. These constraints are crucial to meet memory and inference time requirements, as well as to reduce inference costs. Building small, fast, and private domain-specific language models is a thriving area of research. In this work, we show that a careful pre-training on a *subset* of the public dataset that is guided by the private dataset is crucial to train small language models with differential privacy. On standard benchmarks, small models trained with our new framework achieve state-of-the-art performance. In addition to performance improvements, our results demonstrate that smaller models, through careful pre-training and private fine-tuning, can match the performance of much larger models that do not have access to private data. This underscores the potential of private learning for model compression and enhanced efficiency.

## 1 Introduction

Many papers have shown that deep learning models are vulnerable to attacks aimed at extracting information from the training data (Shokri et al., 2017; Hayes et al., 2019; Carlini et al., 2021; Zhang et al., 2021; Choquette-Choo et al., 2021; Carlini et al., 2023; Matsumoto et al., 2023). A provable path for mitigating such privacy attacks is to train the models with differential privacy (DP) Dwork et al. (2006), a mathematically rigorous notion for quantifying the privacy leakage of a machine learning model. Over the past few years, there has been a rapid progress in our understanding of deep learning with DP, both in terms of computational efficiency (He et al., 2023; Bu et al., 2021; Lee & Kifer, 2021; Subramani et al., 2021; Anil et al., 2022) and privacy-utility trade-off (De et al., 2022; Zhou et al., 2021; Zhu et al., 2020; Golatkar et al., 2022; Sander et al., 2022; Bu et al., 2022a; Panda et al., 2022; Luo et al., 2021; Kairouz et al., 2021; Kurakin et al., 2023).

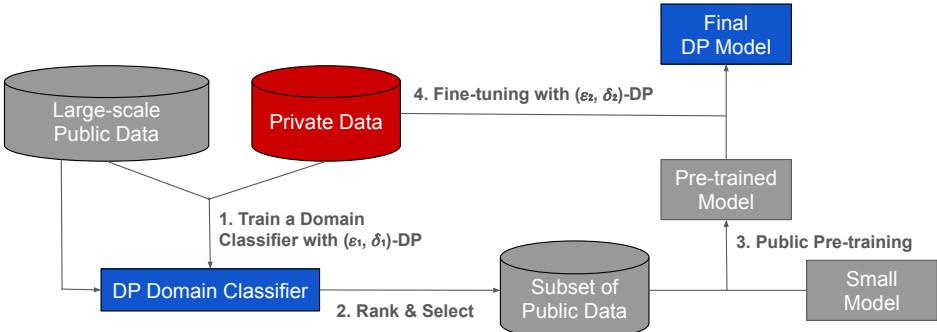

Figure 1: The proposed framework for training a small and domain-specific model with differential privacy (DP). More details on the process of training the domain classifier and the selection of public data can be found in Figure 3. We use the method in Abadi et al. (2016) for training models with DP.

One of the most important findings is that pre-training is crucial for maximizing performance (Li et al., 2022b; Ganesh et al., 2023).

Most of the DP literature mentioned above focus on settings where *inference time* is not a bottleneck and one can deploy models of *any size.* In such a case, existing evidence is that larger models pre-trained on vast amounts of public data perform better when combined with private fine-tuning (Li et al., 2022c; Yu et al., 2022; Mehta et al., 2022). However, there are plenty of applications where the size of the model is restricted by the inference time, e.g., a language model of an email client or a face identification model running in a security system. In such applications, if the inference time is not good then the quality of predictions becomes irrelevant. Further, note also that in both these applications the training data is quite sensitive, and the models should protect the privacy of users. Building small, fast, and private domain specific language models is also a thriving area in industry with several start-ups (MosiacML; ScaleAI). There is also economic motivation as smaller models offer cheaper inference costs.

In this work, we introduce *selective pre-training* as a means to improve DP fine-tuning for small language models. Figure 1 presents an overview of the proposed framework. Specifically, our approach selects a tailored subset of public data, guided by the private data, for model pre-training. This selection process begins with the training of a DP domain classifier, using the private data as positive training samples. The domain classifier is subsequently utilized to rank all public samples, retaining only the top-ranked samples. We conduct extensive experiments to demonstrate the superiority of selective pre-training over standard pre-training. The representative results are presented in Figure 2.

Our main motivation stems from the observation that for a fixed size model, there is an optimal size of pre-training data after which enlarging the pre-training dataset does not further improve downstream performance. This phenomenon is observed in our Figures 2 and 6 and aligns with the previously established scaling laws for pre-training language models (Kaplan et al., 2020; Hoffmann et al., 2022). For example, the results in Figure 2 suggest that for a transformer model with only 21 million parameters, pre-training on a random 15% of OpenWebText (Gokaslan & Cohen, 2019) is no different than pre-training on the full OpenWebText. Therefore, in such a case, selecting a subset of public data that better aligns with the domain of private data is an effective way to fully utilize the constrained capacity of small models.

Our contributions are summarized as follows.

1. We propose selective pre-training as a novel approach for training small, domain-specific language models with DP. We design and implement the first simple and effective DP algorithm for selecting a subset of public data that better aligns with the domain of the private data.

2. We empirically validate the proposed framework using the Enron email dataset (Cohen, 2015) and the GLUE benchmark (Wang et al., 2018). The results from the Enron email dataset (Section 4.1)

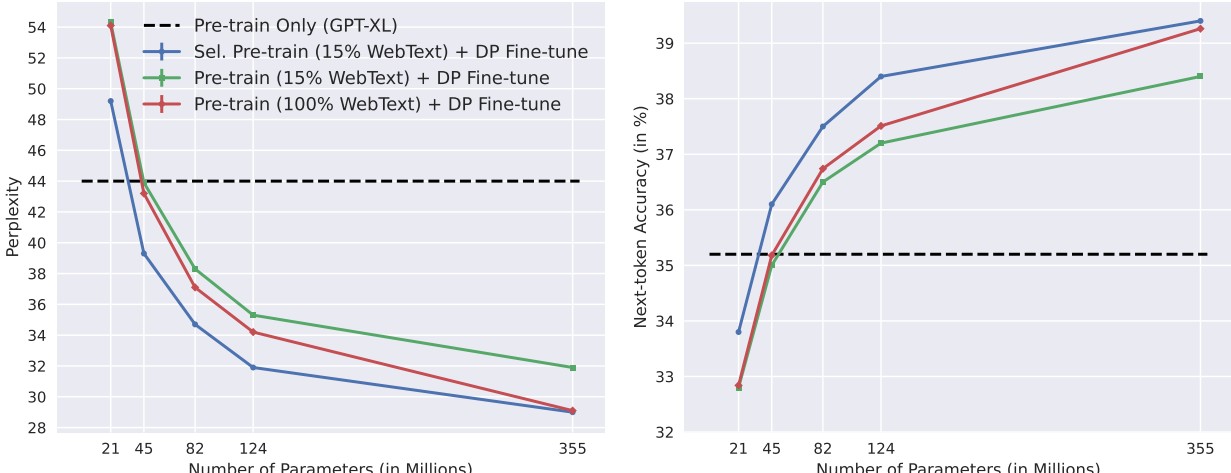

Figure 2: A representative result from our findings. We plot perplexity and top-1 next word accuracy of GPT models on the test set of the Enron email dataset (Cohen, 2015). Overall privacy budget is $(\varepsilon = 7.3, \delta = 1 \times 10^{-7})$. The dashed line shows the zero-shot performance of GPT2-XL with 1.5 billion parameters. The figure shows that our framework yields clear improvements in both perplexity and next-token prediction accuracy, which can significantly improve the overall model behavior.

demonstrate the effectiveness of our framework in real-world scenarios, while the results from the GLUE benchmark (Section 4.2) show its superiority compared to existing baselines.

3. In addition to state-of-the-art DP small models, our experimental results indicate that fine-tuning with DP benefits more significantly from selective pre-training than non-private fine-tuning, as presented in Figures 5 and 7. This underscores the distinctive value of selective pre-training within the DP community.

**Real-world Impact** Our framework was recently used in training an industry grade differentially private text prediction language model that now serves many NLP applications. As text prediction models (on email clients/servers, word processors, etc.) serve billions of queries per hour, the inference cost savings due to the decrease in model size are significant. Further, due to better inference time, online performance metrics, such as the number of predictions accepted by the users, also improve.

### 1.1 Preliminaries

We begin with the formal definition of differential privacy.

**Definition 1** ( $(\epsilon, \delta)$-Differential Privacy (DP) (Dwork et al., 2006)). *A randomized algorithm $\mathcal{A}$ is $(\epsilon, \delta)$-differentially private if for any two neighboring datasets $D$ and $D'$, which differ in exactly one datapoint, and for every subset $\mathcal{S}$ of possible outputs:* $\Pr[\mathcal{A}(D) \in \mathcal{S}] \leq e^{\epsilon} \Pr[\mathcal{A}(D') \in \mathcal{S}] + \delta$.

**Private Deep Learning:** In the context of deep learning, DP guarantees that the trained model *weights* are private with respect to a training dataset, and hence can be released publicly. To train a deep learning model with privacy, the most popular method is to first release the gradients of an optimizer with differential privacy and then update the model with privatized gradients (Song et al., 2013; Bassily et al., 2014; Abadi et al., 2016). We follow the approach in Abadi et al. (2016) to make gradients differentially private. Abadi et al. (2016) augment each minibatch of gradients with per-example gradient clipping and Gaussian noise addition steps. The clipping step ensures that no one user's sample significantly changes the weights of the model and the noise added guarantees that the contribution of a single example is masked.

## 2 Problem Statement and Our Algorithmic Framework

Input to our problem is a private dataset $D_{\mathrm{priv}}$ corresponding to a downstream task $T$, a model $M$ of size $p$, privacy parameters $\epsilon > 0$, $\delta > 0$, and a public dataset $D_{\mathrm{pub}}$. Our goal is to train $M$ on public and

private datasets with the aim of maximizing the downstream performance on the task $T$. The entire process should be $(\epsilon, \delta)$-differentially private with respect to $D_{\mathrm{priv}}$. The constraint on model size is important to compare various algorithms in our setting. In applications, the constraints on model size arise naturally as a consequence of memory and/or inference time requirements.

Our framework for solving the problem consists of the following 3 steps.

1. ***Privacy Preserving Data Selection***: Given $D_{\mathrm{priv}}$, invoke a privacy preserving data selection algorithm $\mathcal{A}_{\mathrm{select}}$ to find a $D'_{\mathrm{pub}} \subseteq D_{\mathrm{pub}}$. The privacy budget for this step is $(\epsilon_1, \delta_1)$.

2. ***Non-Private Pre-training***: Pre-train the model $M$ on $D'_{\mathrm{pub}}$ with a standard pre-training algorithm. This step does not consume any privacy budget.

3. ***Private Fine-tuning***: Fine-tune $M$ on $D_{\mathrm{priv}}$ with a differentially private algorithm $\mathcal{A}_{\mathrm{finetune}}$. The privacy budget for this step is $(\epsilon_2, \delta_2)$.

The non-private pre-training step can be viewed as a post-processing function to $\mathcal{A}_{\mathrm{select}}$ and thus no privacy budget is consumed. The advanced composition theorem of DP (see (Steinke, 2022) for example) guarantees that our framework is $(\epsilon, \delta)$-DP. In our experiments, we use the Privacy Random Variable (PRV) Accountant (Gopi et al., 2021; Ghazi et al., 2022; Koskela et al., 2020). The PRV accountant gives tighter bounds on privacy parameters $\varepsilon$ and $\delta$ than the moments accountant in Abadi et al. (2016). The rest of the paper is devoted to describing the first step of our framework, followed by experiments to verify the effectiveness of our methods on different datasets.

## 3 Privacy Preserving Data Selection

We describe our approach to implementing a privacy-preserving data selection algorithm. We provide a specific implementation of our framework and demonstrate its effectiveness, however, our approach is general and can be combined with other private data selection algorithms.

### 3.1 Our Implementation of Data Selection

Our framework is loosely inspired by the data cleaning framework used in GPT3 and PaLM models Brown et al. (2020); Chowdhery et al. (2022), although motivations are a bit different. The classifiers in Brown et al. (2020); Chowdhery et al. (2022) are trained to filter out noisy documents from datasets. In fact, the source datasets in our paper, i.e., OpenWebText and Wikipedia, are considered positive examples in Brown et al. (2020). Our classifier is trained to recognize examples that are similar to samples in the target data. We initialize the classifier with a pre-trained LM and fine-tune it with differential privacy to predict whether a sentence is sampled from the distribution of the target data. We use the classifier to predict all sentences in the source data and rank them according to confidence scores. Although deep neural networks could be overconfident and need calibration in some applications (Guo et al., 2017; Zhang et al., 2022), not calibrating the outputs does not affect our algorithm because calibration does not change the relative ranking among sentences. We select the top sentences until we reach the target number of pre-training tokens [1]. Figure 3 shows an overview of our implementation.

We create a training set to teach the classifier to recognize a target data distribution. Sentences in the target dataset are labelled as positive. Random samples from the source data are labelled as negative[2]. It has been widely observed that a larger training set helps private learning (Bassily et al., 2014; Tramèr & Boneh, 2021). Therefore we set the number of negative examples as five times larger than the number of positive examples. The privacy cost of training the classifier is accounted in the overall privacy cost.

We run experiments with the Enron Email (Cohen, 2015) as the target and the OpenWebText dataset (Gokaslan & Cohen, 2019) as the source. The classifier is initialized with a 124M GPT model pre-trained on OpenWebText. With a single Tesla A100 GPU, it takes approximately one hour for fine-tuning the domain

---

[1]In our preliminary experiments, we also explored random sampling where the sampling weights are scaled linearly or quadratically with the confidence scores from the domain classifier. We found no statistical difference when compared to using only the top-ranked samples.

[2]This way of assigning negative labels may introduce few noisy labels, since some public samples may closely resemble those from the private dataset. However, given the vast size and diversity of the public dataset, the percentage of noisy labels would be low, as evidenced by the high test F1-score of the trained domain classifiers.

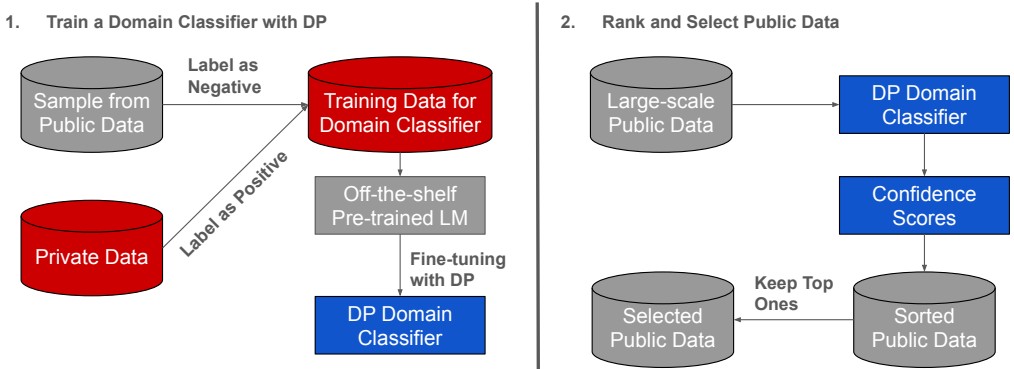

Figure 3: The process of training the domain classifier and the selection of large-scale public data.

classifier. With eight Tesla V100 GPUs, it takes less than two hours for computing the confidence scores for all sequences in OpenWebText. The privacy guarantee is $(0.7, 1 \times 10^{-8})$-DP if we only consider the privacy cost of this step. More implementation details are in Section 4.1. The trained classifier achieves an F1-score of 98.5%. The classifier achieves an F1-score of 92.7% if it is not initialized with a pre-trained LM.

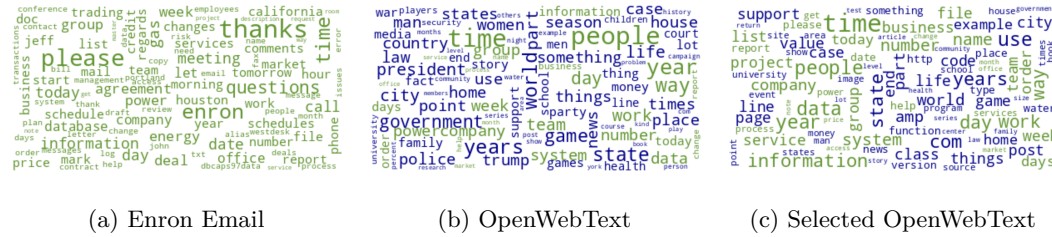

| (a) Enron Email | (b) OpenWebText | (c) Selected OpenWebText |

Figure 4: The 100 most frequent nouns in Enron email, OpenWebText, or a selected subset of OpenWebText (10%). A larger font size indicates that the word appears more frequently. Green words are the 100 most frequent nouns in Enron Email. OpenWebText and selected OpenWebText have 28 and 39 words, respectively, that are among the 100 most frequent nouns in Enron Email.

We use the trained classifier to select 10% of OpenWebText. We plot the word clouds of Enron email, OpenWebText, and the selected subset of OpenWebText (Figure 4), to visually illustrate the dataset selected by our algorithm. The word clouds only show the nouns to exclude common prepositions and verbs. There are 28 nouns which appear in both the top 100 nouns of the Enron email dataset and the top 100 nouns of OpenWebText. The number of overlaps increases to 39 when comparing Enron email with the selected subset of OpenWebText, suggesting the trained domain classifier is an effective tool for data selection. In Appendix B.1, we also present the results of using GLUE (Wang et al., 2018) tasks as the targets and the pre-training corpus of BERT (Devlin et al., 2019) as the source.

## 4 Experimental Evaluation

We evaluate our full framework (Section 2) on language generation and understanding tasks, comparing on datasets that most previous works used (Li et al., 2022c; Yu et al., 2022). The goal here is to empirically verify the main claim made in the introduction: our framework can be used as an effective tool for model compression, and beats the existing baselines in the literature (Mireshghallah et al., 2022). We note that Mireshghallah et al. (2022) did experiments on GLUE benchmark only, and we compare against them in the next section. We begin with the language modeling on the email dataset, which was the motivating example from the real world application.

### 4.1 Implementing the Framework on the Enron Email Dataset

Our first target task is causal language modeling on the Enron email dataset. The dataset contains approximately 0.5 million (M) emails written by employees of the Enron Corporation and is publicly available for research use. We choose this dataset because its distribution closely resembles some private datasets in the real world for which it is hard to find off-the-shelf pre-training data.

#### 4.1.1 Experiment Setup

We briefly describe important parameters of our experimental setup. More details are in Appendix C.

**Target and Source Data** We divide the text into sequences of length 256 and treat each sequence as a datapoint, which constitutes the granularity of our privacy guarantees. Most of the emails in the dataset are shorter than hundred words. In real world applications, it is important to carefully bound the maximum contribution of a single email/user to a training point. There are $\sim$70K sequences in total. We use 80% of them for training and evenly split the rest 20% for validation and testing. The source data is OpenWebText (Gokaslan & Cohen, 2019) which contains $\sim$4 billion tokens. The sequence size for OpenWebText is 512, following the choice in Radford et al. (2018).

**Models** Models in this section are from the GPT family (Radford et al., 2019). We change the number of layers, hidden size, and intermediate size of the fully connected block to get five different model sizes (21M, 45M, 82M, 124M, and 355M). Details of the models and pre-training hyperparameters are in Appendix C. All models are pre-trained with nodes with 8x Nvidia Tesla V100 GPUs.

**Data Selection** We use the algorithm in Section 3 to select 2M sequences from the source data for pre-training. We train the domain classifier for 3 epochs. The baselines include 1) pre-training with 2M random sequences and 2) pre-training with all of OpenWebText.

**Privacy Budget and Hyperparameters** The overall privacy budget is $(7.3, 1 \times 10^{-7})$-DP, similar to previous works on this topic (Li et al., 2022c; Yu et al., 2022). To reduce the privacy cost of hyperparameter tuning (Liu & Talwar, 2019; Papernot & Steinke, 2022; Mohapatra et al., 2022), we follow the findings in previous work to set most of the hyperparameters and only tune the learning rate to adapt to models of different sizes. The hyperparameters for private learning are listed in Table 4 in Appendix C.

#### 4.1.2 Selective Pre-training Provides Clear Gains, Model Efficiency

Figure 2 shows the perplexity and next-word prediction accuracy of different models on the test split of the Enron email dataset. We also present the next-word accuracy and its standard deviation across random seeds in Table 1 in Appendix B.2 as a complementary to Figure 2. It is clear from the figure that our framework improves performance compared to existing techniques.

More significantly, we see that smaller models can match the performance of much larger models; for example, the 82M model using selective pre-training matches the 124M model using normal pre-training. *This shows that the proposed framework can be used to improve the efficiency-utility trade-off of private learning.* We also include the zero-shot performance of the off-the-shelf GPT2-XL model (1.5 billion parameters) in Figure 2. The zero-shot performance of GPT2-XL is worse than the models that have access to private data and are of much smaller size. These findings highlight the importance of private data, which can be loosely treated as high quality data, as well as the importance of privacy-enhancing technologies that facilitate the trustworthy use of such data. Figure 10 in Appendix B.2 also presents the results under different privacy budgets ($\varepsilon$ ranging from 2.3 to 10.9). We observe consistent gains when the selective pre-training framework is used.

#### 4.1.3 Selective Pre-training is More Important for Private Learning

We also fine-tune the models without differential privacy to see whether selective pre-training improves downstream performance in non-private learning. The results are in Figure 5. When using 15% of OpenWebText, selective pre-training still improves the performance of all models though the improvement is smaller compared to the private world. When using 100% of OpenWebText, the benefits of selective pre-training gradually diminish as the model size increases. This suggests that selective pre-training is more important for private learning compared to the case in non-private learning. Our hyperparameters for non-private fine-tuning can be found in Appendix C.

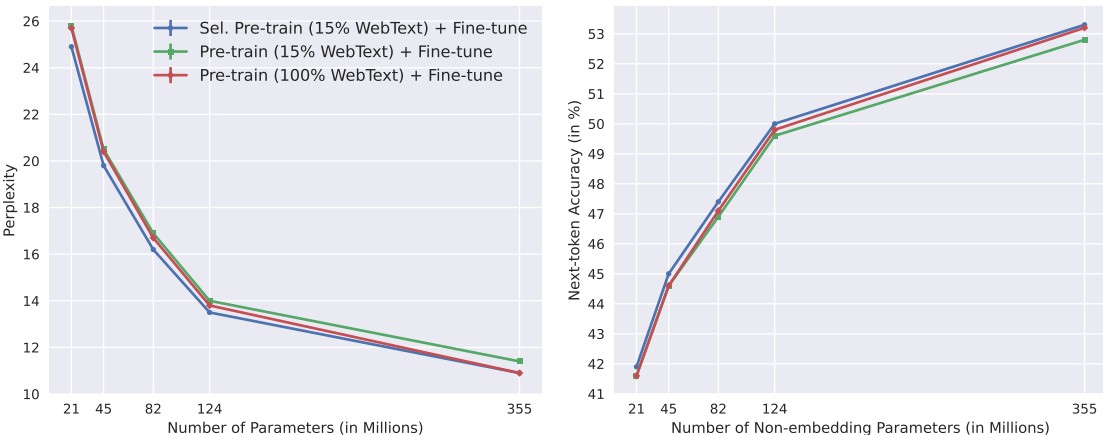

Figure 5: Perplexity and top-1 next word accuracy of GPT models on the test set of the Enron email dataset. The models are trained without DP. Selective pre-training still improves over standard pre-training, however, the improvements are smaller compared to private learning.

## 4.2 Experiments on GLUE

We conduct experiments on the GLUE benchmark (Wang et al., 2018), a common benchmark for fine-tuning language models with DP (Yu et al., 2021; Li et al., 2022c; Bu et al., 2022b). Our results show that selective pre-training also improves DP fine-tuning for language understanding tasks, beating the baselines in Mireshghallah et al. (2022).

### 4.2.1 Experiment Setup

**Target and Source Data** Our target tasks in this section are MNLI and SST-2, which have respectively the largest and smallest number of examples among the four tasks studied in previous work (Yu et al., 2021; Li et al., 2022c; Bu et al., 2022b; Mireshghallah et al., 2022). The numbers of training examples ($N$) in MNLI and SST-2 are 393K and 67K. The source data for GLUE tasks is the pre-training corpus of BERT (Devlin et al., 2019); It consists of a subset of Wikipedia and the entire Bookcorpus. The source dataset has approximately 3.5 billion tokens.

**Model Sizes** We use models from the BERT family (Devlin et al., 2019). We consider four different model sizes (5M, 10M, 25M, and 44M). Details of the models are in Appendix C. Following previous work (Xia et al., 2022), we do not include embedding matrices when computing the number of parameters of BERT models. For text classification tasks, the BERT embedding layer during inference is simply a lookup table.

**Data Selection** For MNLI and SST-2, we experiment with selecting varying numbers of tokens from the source data. The target numbers of pre-training tokens are 20M, 40M, 200M, 400M, 800M, 1200M, and 2000M. More complete implementation details on data selection are in Appendix C.

**Baselines** The baselines include pre-training on randomly selected source data and pre-training on all source data. There are two additional baselines for the 44M model. The first is directly fine-tuning DistillBERT (Sanh et al., 2019) with differential privacy. DistillBERT is distilled from BERT-base on the source data. The second is the best result in Mireshghallah et al. (2022). Mireshghallah et al. (2022) compress a DP fine-tuned BERT-base model using differentially private distillation or pruning. The architecture of the compressed models in Mireshghallah et al. (2022) and Sanh et al. (2019) are of the same architecture as the 44M model. Although our framework is compatible with the techniques in Mireshghallah et al. (2022) and Sanh et al. (2019), we include the two additional baselines to demonstrate that the proposed framework alone is a competitive approach for model compression in private learning.

**Private Learning** We adopt the setup in Mireshghallah et al. (2022). The privacy budget is $(4, 1/10N)$-DP. The hyperparameters for private learning are also documented in Appendix C.

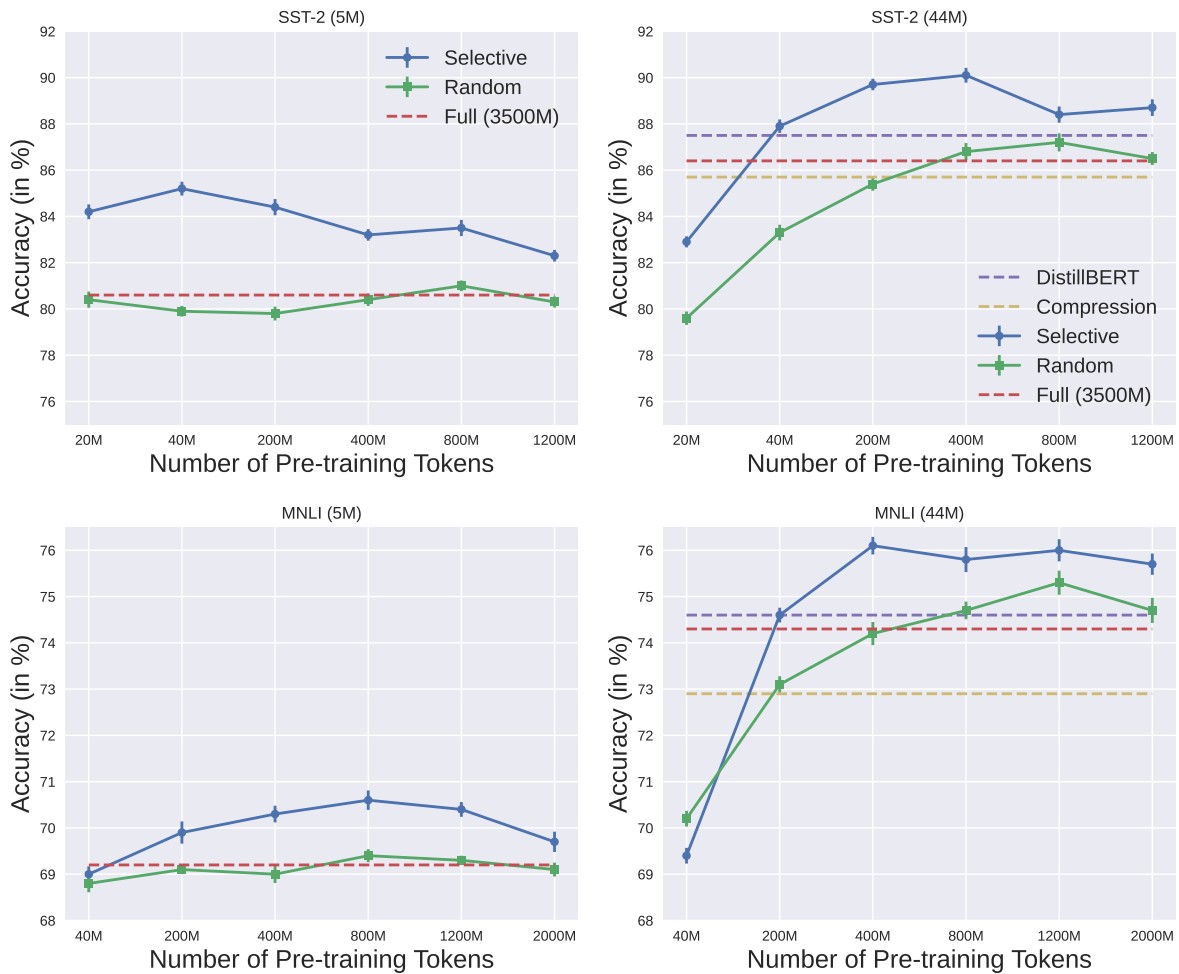

Figure 6: Results of pre-training with various numbers of tokens. The first column shows the results of 5M models and the second column shows the results of 44M models. Selective pre-training outperforms baseline algorithms in most of the cases.

### 4.2.2 Selective Pre-training Outperforms Baselines, Improves Model Efficiency

Figure 6 shows the test accuracy on MNLI and SST-2 after privately fine-tuning models pre-trained with varying numbers of tokens. Our first finding is that, for most of the settings, selective pre-training outperforms all the algorithms examined. On SST-2, selective pre-training achieves accuracy that is 4.6% and 3.7% higher than the accuracy of full pre-training for the 5M and 44M models, respectively. On MNLI, the accuracy improvements are 1.4% and 1.8%, respectively. Our second finding is that, for a model of fixed size, *increasing the number of pre-training tokens does not necessarily lead to better downstream accuracy.* This suggests that there may be an optimal number of pre-training tokens for a given model size (Sorscher et al., 2022; Hoffmann et al., 2022), further emphasizing the need to choose a task-specific subset from a large source data.

Figure 7 shows the test accuracy of models of different sizes. When trained with differential privacy, the 25M model with selective pre-training achieves comparable or better performance than the 44M baseline models, aligning with our observations on the Enron email dataset. The accuracy gains on SST-2 are greater than those achieved on MNLI, likely because MNLI data distribution is relatively closer to Wikipedia corpus; see Appendix B.1 for the word clouds comparison.

## 5 Related Work

Here we discuss closely related prior work. Additional related work is discussed in Appendix A.

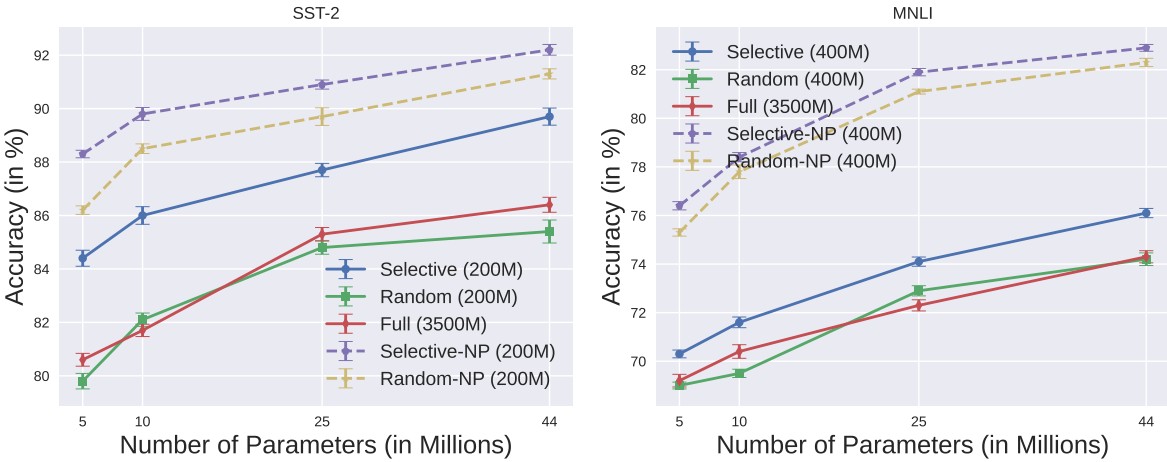

Figure 7: Results of pre-training with different model sizes. The numbers in the brackets are the numbers of tokens used for pre-training. 'NP' denotes that the models are fine-tuned without DP. Selective pre-training consistently improves performance across all settings. The improvements for models trained with DP are larger.

The main scope of this work is to train better small DP language models. This scope aligns closely with the goals of model compression. Mireshghallah et al. (2022) study the implementation of classic model compression techniques, knowledge distillation (Hinton et al., 2015) and model pruning (Han et al., 2015), under DP constraints. While their DP variants achieve promising results on the GLUE benchmark (Wang et al., 2018), they find that the DP constraint reduces the effectiveness of these classic compression techniques compared to the non-private case. In this work, the results in Figure 6 of Section 4.2 suggest that our framework improves upon the results in Mireshghallah et al. (2022) on GLUE, indicating that selective pre-training can serve as an effective new method for training better small DP models.

Hoffmann et al. (2022) find that the optimal number of pre-training tokens scales linearly with the number of model parameters, as shown in their Figure 1 and Figure A3. In this work, we also observe that for a fixed-size model, there is an optimal size of pre-training data, after which further increasing the dataset does not improve downstream performance (see Figure 6 in Section 4.2). However, Hoffmann et al. (2022) focus solely on the quantity of pre-training data, not its quality. In our study, we demonstrate that both the quality, measured by the distributional gap between private and public data, and quantity play an important role in downstream performance. More importantly, Figure 5 and Figure 7 show that DP fine-tuning benefits more from high-quality pre-training compared to non-private fine-tuning, highlighting the unique value of selective pre-training for the privacy-preserving learning community.

Hou et al. (2023) and Gu et al. (2023) study how to privately select an optimal public dataset from an explicitly given list of public datasets. For instance, suppose the private dataset is CIFAR-10, and available public datasets are MNIST, CIFAR100, and ImageNet. The goal is to design a private algorithm to find which of the three public datasets is better suited for private learning on CIFAR-10. In this paper, we explore how to select a subset of a single public dataset on a sample-by-sample basis. Our algorithm does not require any explicit division of public data and runs efficiently on billions of tokens, making it well-suited for finding the right pre-training data for language models. More importantly, our emphasis is not just on model accuracy, but on how pre-training impacts accuracy-vs-model size trade-offs.

# 6 Conclusion and Limitations

## 6.1 Conclusion

This work introduces selective pre-training, a new approach for pre-training language models that are better suited for fine-tuning with DP on private data. The proposed framework pre-trains the models on a selected subset of public data that is better aligned with the domain of the private data. Additionally, the selection of

public data is designed to satisfy DP with respect to the private dataset. Experiments on the Enron email dataset (Cohen, 2015) and GLUE benchmark (Wang et al., 2018) demonstrate that selective pre-training improves the fine-tuning of lightweight language models by clear margins.

## 6.2 Limitations

We provide DP guarantees only for private datasets, not for public ones. In building applications, it is important to consider the privacy risks associated with public data (Tramèr et al., 2022). One potential solution is to enforce differential privacy even during the pre-training stage (Kurakin et al., 2022; Anil et al., 2022).

Our methods pre-train models from scratch, thereby incurring an additional computational cost associated with pre-training compared to prior work that can utilize off-the-shelf pre-trained models (Mireshghallah et al., 2022). However, in real-world applications such as email clients or text editors, the backbone models are queried millions or even billions of times every day. Therefore, the cost of training, being a one-time expense, is a negligible fraction of the accumulated inference cost.

## Broader Impact

Language models, trained with DP or not, are widely used for enhancing the typing experience for users (McMahan et al., 2018; Microsoft, 2020; Xu et al., 2023). In this work, we introduce a new pre-training method designed to improve the DP fine-tuning process of language models. We hope our work can facilitate the adoption of privacy-preserving techniques, providing strong privacy for users while simultaneously reducing deployment costs and further enhancing user experience. However, as previous research indicates, implementing a DP training pipeline can be complex (Tramer et al., 2022). Therefore, we recommend rigorous implementation and thorough auditing for any real-world deployment of the proposed framework.

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

## A   Additional Related Work

For general literature on private deep learning and fine-tuning we refer the readers to (Abadi et al., 2016; He et al., 2023; Kerrigan et al., 2020; Li et al., 2022c; Bu et al., 2021; Lee & Kifer, 2021; Subramani et al., 2021; Anil et al., 2022; Yu et al., 2022; De et al., 2022; Mehta et al., 2022; Yu et al., 2021; Zhu et al., 2020; Sander et al., 2022; Bu et al., 2022a; Panda et al., 2022), and references there in. To the best of our knowledge, no prior work in DP literature has studied selective pre-training from scratch and its impact on the transfer learning abilities of a model. Our work is at the intersection of several related topics, and we give a brief overview of how our work fits into the broader literature on the topic.

**Domain Adaptation and Reducing Distribution Disparity Between Public and Private Data** Public data has been widely used to improve private data analysis (Papernot et al., 2017; Alon et al., 2019; Bassily et al., 2020b;a; Kairouz et al., 2021; Liu et al., 2021a; Zhou et al., 2021; Liu et al., 2021b; Amid et al., 2022; Yang & Cheng, 2022; Li et al., 2022a; Bie et al., 2022). To address the distribution shift between private and public data, a recent line of research explores domain adaption (Wang et al., 2021; Zhang & Gao, 2022) in the context of private learning (Wang et al., 2020; Zheng et al., 2023; Bassily et al., 2023). However, these works are not applicable to our setting due to many reasons, but in particular that we are interested in how pre-training dataset affects the model size. Much of the above literature considers simply the performance of the final model. In the absence of the model size restrictions, for NLP applications, it is well established He et al. (2023) that pre-training on large corpus of text using a large model offers better utility-vs-privacy trade offs.

**Non-Private Data Selection** Automatic data selection and cleaning, along with how the pre-training data impacts the downstream task performance are important problems in deep learning. See Xie et al. (2023); Gururangan et al. (2020); Brown et al. (2020); Chowdhery et al. (2022); Jain et al. (2023); Hernandez et al. (2022); Mindermann et al. (2022); Lee et al. (2022); Coleman et al. (2020) and references there in. Yet, the literature is scarce on the impact of selective pre-training on the model, except the recent concurrent work of Xie et al. (2023). Our work explores these questions in the context of private learning, with an emphasis on how the quality of data affects performance and model size. As a pilot study on designing privacy-preserving data selection algorithms, we use simple classification-based approaches that are easy to privatize and provide a clear illustration of the main messages of the paper. Exploring more sophisticated approaches Xie et al. (2023) for private data selection is an interesting future direction.

## B   More Experiments

### B.1   Results of Data Selection for GLUE Tasks

We plot the word clouds of SST-2/MNLI and (selected) source data to further demonstrate that the distribution of selected data is closer to the distribution of target data. The source data for SST-2 and MNLI is a subset of Wikipedia and the entire Bookcorpus.

The domain classifiers of SST-2 and MNLI are trained the same way as illustrated in Section 3. We select 400M tokens for SST-2 and MNLI, separately. The word clouds of the most frequent 100 nouns are in Figure 8 and 9. We exclude common prepositions and verbs in the word clouds. On SST-2, our selection algorithm improves the number of overlaps between the source data and the target data from 25 to 40. On MNLI, our algorithm improves the number of overlaps from 44 to 51. The results explain our findings in Section 4.2 that selective pre-training yields larger performance improvements on SST-2 than on MNLI.

Table 1: Next word prediction accuracy (in %) of GPT models on the Enron email dataset. The overall privacy budget is $(7.3, 1 \times 10^{-7})$.

| Parameters | 21M | 45M | 82M | 124M | 355M |
|---|---|---|---|---|---|
| Random | $32.8_{\pm 0.02}$ | $35.0_{\pm 0.01}$ | $36.5_{\pm 0.01}$ | $37.2_{\pm 0.02}$ | $38.4_{\pm 0.02}$ |
| Top | $33.8_{\pm 0.03}$ | $36.1_{\pm 0.02}$ | $37.5_{\pm 0.04}$ | $38.4_{\pm 0.02}$ | $39.4_{\pm 0.01}$ |

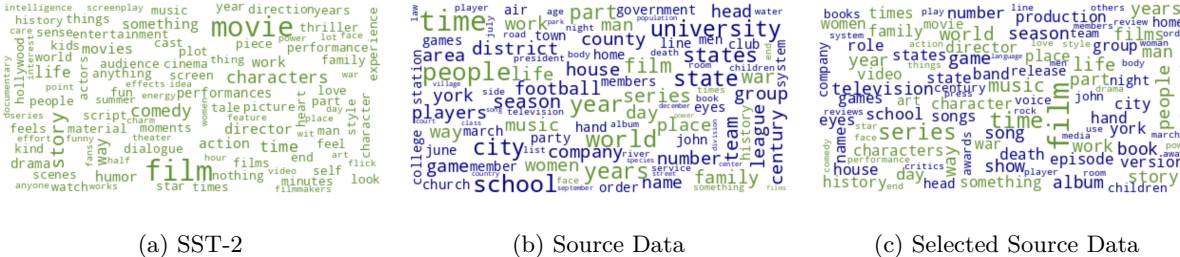

(a) SST-2            (b) Source Data            (c) Selected Source Data

Figure 8: The 100 most frequent nouns in SST-2, the source data, and a selected subset of source data. The source data is Wikipedia and Bookcorpus. Green words are the 100 most frequent nouns in SST-2. The source data and the selected subset have 25 and 40 words, respectively, that are among the 100 most frequent nouns in SST-2.

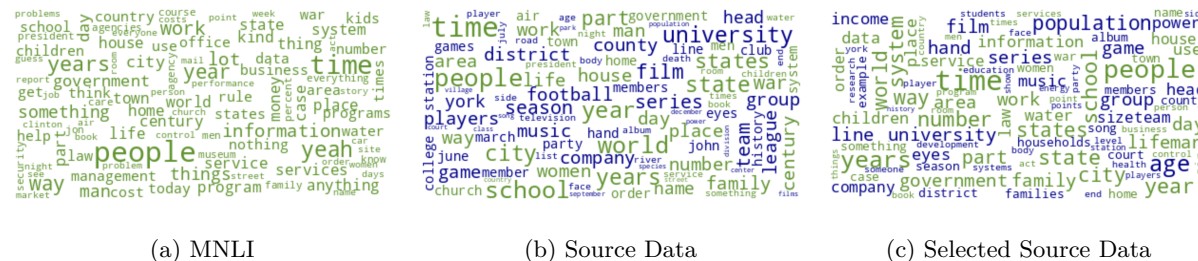

(a) MNLI            (b) Source Data            (c) Selected Source Data

Figure 9: The 100 most frequent nouns in MNLI, the source data, and a selected subset of source data. The source data is Wikipedia and Bookcorpus. Green words are the 100 most frequent nouns in MNLI. The source data and the selected subset have 44 and 51 words, respectively, that are among the 100 most frequent nouns in MNLI.

## B.2  More Experiments on the Enron Email Dataset

Table 1 shows the top-1 next word prediction accuracy on the test split of the Enron email dataset as well as the standard deviation over five random seeds. With selective pre-training, a 82M model achieves an accuracy of 37.5% which is 0.3% higher than the accuracy of a 124M model that is not carefully pre-trained.

We also test selective pre-training under different privacy budgets. Figure 10 presents perplexity and next-word prediction accuracy of 21M and 355M GPT models under a wide range of $\varepsilon$ (ranging from 2.3 to 10.9). We fix the privacy parameter $\delta$ as $1 \times 10^{-7} < 1/10N$. We found that selective pre-training leads to similar improvements across all the choices of $\varepsilon$.

## C  Implementation Details

| Param | 21M | 45M | 82M | 124M | 355M |
|---|---|---|---|---|---|
| $L$ | 4 | 4 | 6 | 12 | 24 |
| $d$ | 312 | 576 | 768 | 768 | 1024 |
| $d_{FFN}$ | 1248 | 2304 | 3072 | 3072 | 4096 |

| Param | 5M | 10M | 25M | 44M |
|---|---|---|---|---|
| $L$ | 4 | 6 | 6 | 6 |
| $d$ | 312 | 384 | 576 | 768 |
| $d_{FFN}$ | 1200 | 1200 | 2304 | 3072 |

Table 2: Architecture hyperparameters of the models for the Enron email dataset.

Table 3: Architecture hyperparameters of the models for GLUE tasks.

This section expands on the implementation details that are omitted from the main text due to space constraints.

**Details of the Models**  Let $L$, $d$, and $d_{FFN}$ be the number of layers, hidden size, and intermediate size of the fully connected block, respectively. We change $L$, $d$, $d_{FFN}$ to get different model sizes. Other architecture

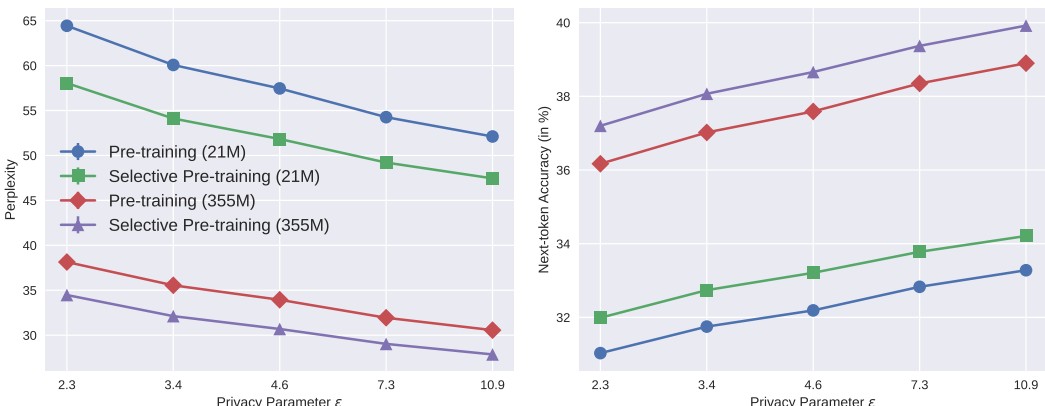

Figure 10: Perplexity and top-1 next-word accuracy on the Enron email dataset. We consider a wide range of $\varepsilon$ (ranging from 2.3 to 10.9). The numbers in brackets are the number of model parameters. The privacy parameter $\delta$ is $1 \times 10^{-7}$. Selective pre-training yields consistent gains across all $\varepsilon$ evaluated.

hyperparameters are the same as those in Devlin et al. (2019) and Radford et al. (2019). Table 2 and 3 show the model details for the Enron email dataset and GLUE tasks, respectively.

**Data Selection for Enron Email**   The text in OpenWebText is also divided into sequences of length 256. To construct the training set of the domain classifier, we randomly sample $5N$ sequences from the source data as negative samples, and use all $N$ sequences in the target dataset as positive samples. As a result, the training set of the domain classifier is 6 times larger than the target data. This significantly reduces the privacy cost of training the domain classifier because the probability of a target example being sampled becomes 6 times smaller. We initialize the domain classifier with an 82M GPT model pre-trained on OpenWebText and fine-tune it with DP-Adam on the constructed training set.

**Data Selection for GLUE Tasks**   Because the positive examples in SST-2 and MNLI are natural sentences instead of sequences of fixed length, we sample natural sentences in the source data as negative examples for training the domain classifier. The domain classifier is initialized with BERT-base. In MNLI, a single example contains two natural sentences, i.e., a premise and a hypothesis. In this case, only one of the two sentences is chosen randomly as a positive example. The number of negative examples is also $5N$.

The pre-training sequences in Devlin et al. (2019) are of a fixed length. Each sequence may consist of several natural sentences. To get the ranking score of a sequence, we first break a fixed-length sequence into natural sentences and use the domain classifier to predict those sentences. The maximum confidence of the sentences is used as the ranking score for the sequence.

Table 4: Hyperparameters for private fine-tuning. We use $N$ to denote the size of the target dataset.

| Pre-training Method | Standard | Selective |
|---|---|---|
| Noise multiplier (Enron) | 1.00 | 1.03 |
| Noise multiplier (SST-2) | 1.36 | 1.38 |
| Noise multiplier (MNLI) | 1.44 | 1.46 |
| Train steps (domain classifier) | N/A | 100 |
| Train steps (target task) | | $[150, 500, 1000]$ |
| Clipping norm | | 1 |
| Learning rate | | $[\text{1e-4, 5e-4, 1e-3, 3e-3}]$ |
| Weight decay | | 0 |
| Batchsize | | $\lfloor 0.03N \rfloor$ |
| Privacy budget | | $(7.3, 1 \times 10^{-7})$ for Enron; $(4, 1/10N)$ for GLUE |

**Hyperparameters For Pre-training**   The pre-training process uses common hyperparameters in the literature. For pre-training models from the BERT family, we follow the hyperparameters in Devlin et al. (2019). The hyperparameters for pre-training models from the GPT family are as follows. We use a dropout probability of 0.1 and a weight decay of 0.01. The $\beta_1$ and $\beta_2$ of Adam are 0.9 and 0.999, respectively. All models are pre-trained from scratch for 100K iterations with a batch size of 128. The initial learning rate is $5 \times 10^{-4}$ and follows a linear decay schedule.

**Hyperparameters For Private Fine-tuning**   We follow the findings in previous work to set most of the hyperparameters (Li et al., 2022c; Mireshghallah et al., 2022). We additionally tune the learning rate to adapt to the various model sizes we studied. Table 4 summarizes the hyperparameters for private learning. We use the parameter-efficient fine-tuning algorithm LoRA (Hu et al., 2022) to improve the efficiency of the DP fine-tuning of GPT models Yu et al. (2022); Kurakin et al. (2023). We do not use LoRA for the DP fine-tuning of BERT models to get a fair comparison to Mireshghallah et al. (2022). For a given set of hyperparameters, we use the PRV accountant to get the noise multiplier of DP-Adam. If we use selective pre-training, then the noise multiplier is slightly larger because we need to account for the privacy cost of training a domain classifier. We repeat each private fine-tuning experiment 5 and 3 times with different random seeds for the Enron email dataset and GLUE, respectively.

