# OpenReview forum: "Selective Pre-training for Private Fine-tuning"
_TMLR — Accepted by TMLR_

### Review · Reviewer_9H2K · 2024-03-15

**Summary Of Contributions:**

The paper's key contribution is a strategy for pre-training small language models with differential privacy. The strategy involves using the private dataset to guide the selection of a subset of the public dataset, which is then used for pre-training. The authors argue that this approach is essential for achieving good results when training these models with differential privacy.

**Audience:**

Yes

**Claims And Evidence:**

No

**Requested Changes:**

See weakness

**Strengths And Weaknesses:**

# Strengths

The paper tries to use data selection techniques to improve downstream task performance under private scenario. The paper makes an effort to provide the experiment of pretraining from scratch, which can take time.

# Weaknesses

Firstly, I think there are several major issues of the figure in the paper; figure 1 is quite rough, which lack the meanings of terms such as "dp data selection" and "private finetuning." It's crucial for the authors to clarify these terms within the figure to enhance its clarity and utility; For figure 2, why "pretrain-only" is a flat line, for different size of the pretrained model, they are also expected to have different perplexity; Regarding Figure 3, there appears to be a significant flaw in the implication that increasing the quantity of data alone enhances language model performance. This overlooks the quality aspect; simply adding low-quality data may not improve model performance.

Additionally, I fail to grasp the connection between privacy-preserving learning and high-quality training data, which the author states in the paragraph.
Despite the authors' assertion that their method serves as a tool for model compression, I don't observe any evident model compression techniques in the paper. As I understand it, the approach seems to involve taking a subset of data and integrating it into the pretraining stage only.
Additionally, the claim of being state-of-the-art lacks substantiation, as there's no direct comparison in the experimental figures. Moreover, the cited paper was published in 2022, prompting questions about the results of other data selection methods introduced since then.

A major concern is the lack of a clear main contribution in the paper. The absence of a conclusion section further exacerbates this issue. Even after reviewing the abstract and introduction, I still lack a comprehensive understanding of the paper's primary objective.
Overall, I believe the paper falls below the journal's threshold due to its lack of novelty and shortcomings in both writing and experimental design. Careful revision is required to address these issues effectively.

---

> ### Author Response · Authors · 2024-04-15
> **Author Response (1/2)**
>
> Thank you for your comments and for taking the time to review our paper. We have revised our submission accordingly and clarified any misunderstandings. We have added a blue mark '[Revised]' before each revised paragraph and figure.
>
>
> **1.** A major concern is the lack of a clear main contribution in the paper. The absence of a conclusion section further exacerbates this issue.
>
> In the revised version, we have reorganized our contributions in the introduction and added a conclusion section. We are happy to discuss further if there are still concerns about clarity.
>
> **2.** There are several major issues with the figures.
>
> - Figure 1 is quite rough, some terms are not clarified.
>
> We have revised Figure 1 to make it more informative. Figure 1 now includes an outlined process of training the domain classifier and the selection of public data. In the caption of Figure 1, we have added a reference to the source of the private fine-tuning algorithm.
>
> - For Figure 2, why "Pre-train Only (GPT-XL)" is a flat line.
>
> This is because it shows the zero-shot performance of a fixed model. In the caption of Figure 2 of the initial submission, we wrote that “The dashed line shows the zero-shot performance of GPT2-XL with 1.5 billion parameters.”.
>
> - Regarding Figure 3, there appears to be a significant flaw in the implication that increasing the quantity of data alone enhances language model performance.
>
> We thank the reviewer for raising the question. In Figure 3 of the initial submission, we implicitly assumed that the data quality axis's origin point represents the quality of standard pre-training data, and the model performance could be improved by improving the data quality through selective pre-training, which is the main focus of this work.
>
> In the revised version, we have rewritten and moved the discussion to the Related Work section (presented below) and removed the original Figure 3.
>
> “Hoffmann et al. (2022) [1] find that the optimal number of pre-training tokens scales linearly with the number of model parameters, as shown in their Figure 1 and Figure A3. In this work, we also observe that for a fixed-size model, there is an optimal size of pre-training data, after which further increasing the dataset does not improve downstream performance (see Figure 6 in Section 4.2). However, Hoffmann et al. (2022) focus solely on the quantity of pre-training data, not its quality. In our study, we demonstrate that both the quality, measured by the distributional gap between private and public data, and quantity play an important role in downstream performance. More importantly, Figure 5 and Figure 7 show that DP fine-tuning benefits more from high-quality pre-training compared to non-private fine-tuning, highlighting the unique value of selective pre-training for the privacy-preserving learning community.”
>
>
> **3.** The connection between privacy-preserving learning and high-quality training data is unclear.
>
> For pre-training purposes, 'high-quality data' refers to a subset of public data whose distribution closely aligns with that of the private data. The process of finding such a high-quality subset, guided by the private dataset, should preserve privacy. Therefore, in this work, we introduce the first differentially private (DP) algorithm designed for this purpose.
>
> **4.** Despite the authors' assertion that their method serves as a tool for model compression, I don't observe any evident model compression techniques in the paper.
>
> In our view, model compression broadly refers to techniques aimed at enhancing model efficiency. Therefore, our framework, which is designed to train small DP language models, can be considered one approach for model compression.
>
> In the revised version, we have moved the discussion on model compression to the Related Work section (presented below).
>
> “The main scope of this work is to train better small DP language models. This scope aligns closely with the goals of model compression. Mireshghallah et al. (2022) study the implementation of classic model compression techniques, knowledge distillation (Hinton et al., 2015) and model pruning (Han et al., 2015), under DP constraints. While their DP variants achieve promising results on the GLUE benchmark (Wang et al., 2018), they find that the DP constraint reduces the effectiveness of these classic compression techniques compared to the non-private case. In this work, the results shown in Figure 6 of Section 4.2 suggest that our framework improves upon the results in Mireshghallah et al. (2022) on GLUE, indicating that selective pre-training can serve as an effective new method for training better small DP models.”

---

> ### Author Response · Authors · 2024-04-15
> **Author Response (2/2)**
>
> **5.** The claim of being state-of-the-art lacks substantiation, as there's no direct comparison in the experimental figures. Moreover, the cited paper was published in 2022, prompting questions about the results of other data selection methods introduced since then.
>
> In Figure 7 of the original submission (Figure 6 in the revised version), the dashed 'Compression' line shows the best results in Mireshghallah et al. (2022). The comparison in Figure 7 demonstrates the superiority of our framework compared to that of Mireshghallah et al. (2022).
>
> We kindly note that Mireshghallah et al. (2022) do not employ any data selection method. They study DP variants of both knowledge distillation and model pruning techniques. This submission is the first to explore pre-training data selection under DP constraints. Although the non-private literature offers methods for selecting pre-training data, as discussed in Appendix A, these methods do not provide a DP guarantee for private data. For example, Xie et al. (2023) [2] select pre-training data based on the unigrams or bigrams of downstream data; however, the use of unigrams and bigrams does not provide privacy guarantee to the downstream data.
>
>
> **References**:
>
> [1]: Training Compute-Optimal Large Language Models, 2022, https://arxiv.org/abs/2203.15556.
>
> [2]: Data Selection for Language Models via Importance Resampling, NeurIPS 2023, https://arxiv.org/abs/2302.03169.

---

> ### Author Response · Authors · 2024-04-22
>
> Dear Reviewer 9H2K,
>
> Thank you again for your review. We have uploaded a revised version of our manuscript to address your concerns. Please refer to our `Author Response` for details. As the author-reviewer discussion period is nearing its end, we would greatly appreciate if you could take a look at our response and the revised manuscript. We are also ready to answer any remaining questions you might have.
>
> Authors of Paper 2278

---

### Review · Reviewer_88tk · 2024-04-04

**Summary Of Contributions:**

The paper proposes a method to select public samples for the pre-training LLMs for private in-domain-data-assisted private training of LLMs. This is done such that part of the privacy budget is spend on training a binary classifier to detect whether the sample was in the private domain or not (randomly selected public samples are labelled negative, private samples positive for training the classifier). Then, pre-training samples are sought from the public data such that the highest-confidence samples (scores obtained using the classifier) are included for pre-training. Several interesting conclusions can be drawn from the experimental results, such as that this selective pre-training benefits the private models more than it does non-private models and that with selective pre-training much smaller models can be used. Also, the optimal number of pre-training samples to choose seems to be a function of the model size, smaller models have smaller optimal number of pre-training samples.

**Audience:**

Yes

**Broader Impact Concerns:**

Considering the method has been implemented for an industry scale differentially private text prediction LM, I think it would be suitable to have a broader impact section in the paper. What are the ethical implications, the pros and cons of having a DP model in the email server and so on.

**Claims And Evidence:**

Yes

**Requested Changes:**

The scaling laws are shortly mentioned in the beginning and also very shortly in the end in the section on open questions: "Our work also touches upon scaling laws..". It feels a bit vague the way it is written, could you elaborate on that? How could those laws (e.g., number of pre-training samples as a function of the model size) look like and what would possibly be the reasoning behind those laws?

I have the following 'open question' in mind, perhaps you could address this: Considering different fine-tuning / pertaining strategies for LLMs, what do you think would be an optimal approach in case you need to retrain the model? I suppose it would help to reselect source samples and retrain the DP selecion model as well in this method? Could you comment which adaptors / fine-tuning strategies might be the best in case you want to retrain?
Related to this: the following paper considers different approaches for fine-tuning LLMs (perhaps should be cited?) :

Kurakin, A., Ponomareva, N., Syed, U., MacDermed, L., & Terzis, A. (2023). Harnessing large-language models to generate private synthetic text. arXiv preprint arXiv:2306.01684.

Also, what do you think how would the choice of the fine-tuning / adaptation strategy (see e.g. the paper by Kurakin et al. above) affect the 'scaling laws' i.e. whether more or less pre-trainining samples would be beneficial? Or do you think it would simply be a function of the number of trainable parameters?

Did you try different strategies for choosing the pre-training samples using the privately trained selection model? You seem to simply rank the samples and take the ones with highest scores. Perhaps some randomized approach could be an alternative?

Typo: end of page 9, "recently" twice in a sentence.

**Strengths And Weaknesses:**

Strenghts:

- A very well written paper, carefully designed replicable experiments that clearly indicate that the approach has its benefits.

- Simple and useful idea.

Weaknesses:

- The contribution is almost totally based on the experimental results, no theoretical/technical results to support the observations and conclusions.

---

> ### Author Response · Authors · 2024-04-15
> **Author Response (1/2)**
>
> Thank you for the detailed comments and suggestions. Below are our responses to each point. We have uploaded a revised version addressing the reviewers' comments. Each revised paragraph and figure are marked with a blue '[Revised]' tag.
>
> **1.** The scaling laws are shortly mentioned in the beginning and also very shortly in the end. It feels a bit vague the way it is written, could you elaborate on that? How could those laws look like and what would possibly be the reasoning behind those laws?
>
> In the revised version, we have added a discussion on scaling laws in the Related Work section (presented below). In the discussion, we elaborate on why this work touches upon scaling laws and explain how the scaling laws look like.
>
> “Hoffmann et al. (2022) find that the optimal number of pre-training tokens scales linearly with the number of model parameters, as shown in their Figure 1 and Figure A3. In this work, we also observe that for a fixed-size model, there is an optimal size of pre-training data, after which further increasing the dataset does not improve downstream performance (see Figure 6 in Section 4.2). However, Hoffmann et al. (2022) focus solely on the quantity of pre-training data, not its quality. In our study, we demonstrate that both the quality, measured by the distributional gap between private and public data, and quantity play an important role in downstream performance. More importantly, Figure 5 and Figure 7 show that DP fine-tuning benefits more from high-quality pre-training compared to non-private fine-tuning, highlighting the unique value of selective pre-training for the privacy-preserving learning community.”
>
> In the introduction, we also briefly mentioned that the reasoning behind such laws could be related to the constrained capacity of fixed size models. However, we acknowledge that the community still lacks a deep understanding of the reasons behind these scaling laws, and that providing such understanding remains an intriguing research direction.
>
> **2.** Considering different fine-tuning / pertaining strategies for LLMs, what do you think would be an optimal approach in case you need to retrain the model?
>
> For selective pre-training, if the source dataset remains unchanged and only the pre-training strategies, such as optimizers, are modified, we do believe the domain classifier can be reused.
>
> **3.** Could you comment which adaptors / fine-tuning strategies might be the best in case you want to retrain? & What do you think how would the choice of the fine-tuning / adaptation strategy (see e.g. the paper by Kurakin et al. above) affect the 'scaling laws' i.e. whether more or less pre-trainining samples would be beneficial?
>
> We thank the reviewer for pointing out the work of Kurakin et al. (2023) to us. We have added the citation.
>
> Regarding the choice of fine-tuning strategies, both DP full fine-tuning and DP parameter-efficient fine-tuning have been found effective for small to medium-sized language models, according to Yu et al. (2022) [1] and Li et al. (2022) [2]. For instance, Table 2 in Li et al. (2022) shows that these two methods result in similar performance when fine-tuning GPT-2. However, according to Table 1 of Kurakin et al. (2023), parameter-efficient fine-tuning substantially outperforms full fine-tuning when applied to a LLM with 8 billion parameters.
>
> Regarding how the choice of fine-tuning strategy affects the scaling laws. In our initial submission, we explored two fine-tuning strategies and found that the optimal number of pre-training tokens depends on the model size for both cases. As detailed in the last paragraph of Appendix C, we use full fine-tuning for BERT models and parameter-efficient fine-tuning for GPT models. We use full fine-tuning for BERT so that our results were comparable with those reported in Mireshghallah et al. (2022).

---

> > ### Author Response · Authors · 2024-04-15
> > **Author Response (2/2)**
> >
> > **4.** Did you try different strategies for choosing the pre-training samples using the privately trained selection model? Perhaps some randomized approach could be an alternative?
> >
> > In our preliminary experiments, we also explored random sampling where the sampling weights are scaled linearly or quadratically with the confidence scores from the domain classifier. We found no statistical difference when compared to using only the top-ranked samples. Therefore, given the high cost of pre-training experiments, we only kept the top-ranked samples for our main experiments. In the revised version, we have clarified this in Section 3.1.
> >
> > **5.** Considering the method has been implemented for an industry scale differentially private text prediction LM, I think it would be suitable to have a broader impact section in the paper.
> >
> > We have added a broader impact section that includes the following discussion.
> >
> > "Language models, trained with DP or not, are widely used for enhancing the typing experience for users [3, 4, 5]. In this work, we introduce a new pre-training method designed to improve the DP fine-tuning process of language models. We hope our work can facilitate the adoption of privacy-preserving techniques, providing strong privacy for users while simultaneously reducing deployment costs and further enhancing user experience. However, as previous research indicates, implementing a DP training pipeline can be complex [6]. Thus, we recommend rigorous implementation and thorough auditing for any real-world deployment of the proposed framework."
> >
> > **6.** Typo
> >
> > Thank you for pointing this out. We have fixed it in the revision.
> >
> > **References**:
> >
> > [1]: Differentially Private Fine-tuning of Language Models, ICLR 2022, https://arxiv.org/abs/2110.06500.
> >
> > [2]: Large Language Models Can Be Strong Differentially Private Learners, ICLR 2022, https://arxiv.org/abs/2110.05679.
> >
> > [3]: Learning Differentially Private Recurrent Language Models, ICLR 2018, https://arxiv.org/abs/1710.06963.
> >
> > [4]: Assistive AI Makes Replying Easier, 2020, https://www.microsoft.com/en-us/research/group/msai/articles/assistive-ai-makes-replying-easier-2/.
> >
> > [5]: Federated Learning of Gboard Language Models with Differential Privacy, ACL 2023, https://arxiv.org/abs/2305.18465.
> >
> > [6]: Debugging Differential Privacy: A case Study for Privacy Auditing, 2022, https://arxiv.org/abs/2202.12219.

---

> > > ### Comment · Reviewer_88tk · 2024-04-26
> > >
> > > Thank you for the replies! The revision addresses my concerns. Thanks especially for the new paragraph on scaling laws and for the broader impact section.

---

> > > > ### Author Response · Authors · 2024-04-27
> > > >
> > > > Thank you for acknowledging our response!

---

### Review · Reviewer_n8f8 · 2024-04-09

**Summary Of Contributions:**

In this work, the authors show that pre-training on a subset of the public dataset, selected under the guidance of a private dataset, can improve the performance of the small, domain-specific model in the differential privacy fine-tuning scenario, superior to the model pre-trained on the whole public dataset.

A public data selection algorithm is proposed to select the subset of the public dataset under the guidance of the private dataset. Solid experiments show that pre-training a small model on the selected public dataset can improve the performance of the DP-SGD fine-tuning with the private dataset.

**Audience:**

Yes

**Claims And Evidence:**

Yes

**Requested Changes:**

See comments above.

**Strengths And Weaknesses:**

Strengths:

- Simple and effective method that not only has great potential practical value but also adds to our understanding of scaling laws.

- Solid experimental results that well support the claims, showing the trade-off between both privacy-utility and efficiency-utility

Weaknesses:

- The data selection algorithm is quite simple and all the randomly used source data are labeled as negative samples. Although this work is an excellent start in the differential private data-selection topic, the proposed algorithm is certainly not the optimal way to select source training samples since potential positive data may be labeled as negative.

- On page 5, the author mentioned A100 GPU, but in the later section, it is claimed that all the models are trained on V100 GPU.

---

> ### Author Response · Authors · 2024-04-15
> **Author Response**
>
> Thank you for the insightful comments. Please find our response below. We have uploaded a revised version addressing comments from all reviewers. Each revised paragraph and figure are marked with a blue '[Revised]' tag.
>
> **1.** All the randomly used source data are labeled as negative samples. This is certainly not optimal since potential positive data may be labeled as negative.
>
> We thank the reviewer for raising this question. We acknowledge that the current way of assigning negative labels may produce noisy labels. In the revised version, we have added the following discussion to Section 3.1.
> “This way of assigning negative labels may introduce few noisy labels, since some public samples may closely resemble those from the private dataset. However, given the vast size and diversity of the public dataset, the percentage of noisy labels would be low, as evidenced by the high test F1-score of the trained domain classifiers.”
>
> One advantage of this simple method is that it is independent of the private dataset, thereby ensuring that creating the domain classifier's training set does not incur additional privacy costs.
>
> **2.** On page 5, the author mentioned A100 GPU, but in the later section, it is claimed that all the models are trained on V100 GPU.
>
> In section 4.1.1 of the initial submission, we wrote that “All models are **pre-trained** with nodes with 8x Nvidia Tesla V100 GPUs.”. The use of a single A100 GPU, as mentioned on page 5, is for some fine-tuning experiments. We have clarified this in the revised version.

---

### Decision · Action_Editor_gxUW · 2024-05-28

**Recommendation:** Accept as is

**Comment:**

A majority of reviewers agree that this is a nice addition to the literature on private language models and that the work provides a interesting perspective on scaling laws. One reviewer was not in favor of acceptance, raising concerns regarding novelty (not part of TMLR's criteria) and shortcomings in writing and experimental design (not shared by the other two reviewers).

**Audience:**

Private fine-tuning of (language) models is a topic highly relevant for the TMLR audience.

**Claims And Evidence:**

The paper presents an approach where private data is used to (privately) select relevant public datasets for pre-training a small language model. Overall, the idea is interesting and works well in practice, as supported by experiments.